# Modulation of pulsatile GnRH dynamics across the ovarian cycle via changes in the network excitability and basal activity of the arcuate kisspeptin network

**Margaritis Voliotis[1]\*[†], Xiao Feng Li[2†], Ross Alexander De Burgh[2†], Geffen Lass[2], Deyana Ivanova[2], Caitlin McIntyre[2], Kevin O'Byrne[2], Krasimira Tsaneva-Atanasova[1]**

[1]Department of Mathematics and Living Systems Institute, College of Engineering, Mathematics and Physical Sciences, University of Exeter, Exeter, United Kingdom; [2]Department of Women and Children's Health, School of Life Course Sciences, King's College London, London, United Kingdom

**\*For correspondence:** m.voliotis@exeter.ac.uk

[†]These authors contributed equally to this work

**Abstract** Pulsatile GnRH release is essential for normal reproductive function. Kisspeptin secreting neurons found in the arcuate nucleus, known as KNDy neurons for co-expressing neuro-kinin B, and dynorphin, drive pulsatile GnRH release. Furthermore, gonadal steroids regulate GnRH pulsatile dynamics across the ovarian cycle by altering KNDy neurons' signalling properties. However, the precise mechanism of regulation remains mostly unknown. To better understand these mechanisms, we start by perturbing the KNDy system at different stages of the estrous cycle using optogenetics. We find that optogenetic stimulation of KNDy neurons stimulates pulsatile GnRH/LH secretion in estrous mice but inhibits it in diestrous mice. These in vivo results in combination with mathematical modelling suggest that the transition between estrus and diestrus is underpinned by well-orchestrated changes in neuropeptide signalling and in the excitability of the KNDy population controlled via glutamate signalling. Guided by model predictions, we show that blocking glutamate signalling in diestrous animals inhibits LH pulses, and that optic stimulation of the KNDy population mitigates this inhibition. In estrous mice, disruption of glutamate signalling inhibits pulses generated via sustained low-frequency optic stimulation of the KNDy population, supporting the idea that the level of network excitability is critical for pulse generation. Our results reconcile previous puzzling findings regarding the estradiol-dependent effect that several neuromodulators have on the GnRH pulse generator dynamics. Therefore, we anticipate our model to be a cornerstone for a more quantitative understanding of the pathways via which gonadal steroids regulate GnRH pulse generator dynamics. Finally, our results could inform useful repurposing of drugs targeting the glutamate system in reproductive therapy.

## Editor's evaluation

Successful reproduction requires luteinizing hormone that is secreted from the pituitary gland in pulses which vary over the female reproductive cycle. The pulses arise by patterned secretion of a hypothalamic 'releasing factor', the secretion of which is itself governed by a population of hypothalamic neurons that express the neuropeptide kisspeptin. The paper by Voliotis et al. combines novel experimental evidence from transgenic mice with an elegant mathematical model to analyze how the kisspeptin neurons generate the varying pulsatile patterns. Testing this model in a way that will either confirm or refute it, may prove challenging.

## Introduction

The dynamics of gonadotropin-releasing hormone (GnRH) secretion is critical for reproductive health. In female animals, GnRH secretion is tightly regulated across the ovarian cycle. Pulsatile secretion dominates most of the cycle, with frequency and amplitude modulated by the ovarian steroid feedback. Positive feedback from increasing estradiol levels triggers a preovulatory surge of GnRH/LH secretion (*Christian and Moenter, 2010*). Furthermore, there is ample evidence that ARC kisspeptin neurons are prime mediators of the ovarian steroid feedback on the pulsatile dynamics of GnRH/LH secretion (*McQuillan et al., 2019*), although the mechanisms remain unclear.

In vitro studies have shown that gonadal steroids have a dramatic effect on the electrophysiology of ARC kisspeptin neurons. For instance, spontaneous firing activity of ARC kisspeptin neurons from castrated mice appears elevated compared to intact animals (*Ruka et al., 2016*) and estradiol replacement attenuates ARC kisspeptin neuron activity in gonadectomised animals (*Ruka et al., 2016*; *Wang et al., 2018*). More recently, fibre photometry data from female mice show that the ARC kisspeptin neuronal population (KNDy network) pulses at a relatively constant frequency throughout the ovarian cycle apart from the estrous phase where the frequency is dramatically reduced (*McQuillan et al., 2019*). This slowdown of LH frequency is thought to be a direct consequence of the increasing progesterone levels associated with ovulation (*McQuillan et al., 2019*), although studies using the rhesus monkey show that raising pre-ovulatory estrogen levels are also important (*O'Byrne et al., 1991*). Studies in sheep indicate the inhibitory effects of progesterone are mediated through increased dynorphin signalling (*Goodman et al., 2011*; *Moore et al., 2018*); however, this is less clear in mice where ovarian steroids have a negative effect on Dyn mRNA levels (*Navarro et al., 2009*).

Perplexing is also the differential effect of various neuromodulators on LH secretion depending on the gonadal steroid background. For instance, N-methyl-D-aspartate (NMDA) robustly inhibits LH pulses in the ovariectomised monkey, whereas in the presence of estradiol this effect is reversed, and NMDA stimulates LH secretion (*Reyes et al., 1990*; *Reyes et al., 1991*). Similar reversal of action on LH dynamics depending on the underlying ovarian steroid milieu has been also documented for other neurotransmitter and neuropeptides in other species (*Kalra and Kalra, 1983*; *Brann and Mahesh, 1992*; *Arias et al., 1993*; *Bonavera et al., 1994*; *Scorticati et al., 2004*) and highlights the complex mechanisms underlying the modulation the GnRH pulse generator by gonadal steroids.

Here, using mathematical modelling along with optogenetic stimulation of ARC kisspeptin neurons, we embark to understand how the dynamics of the pulse generator are modulated across the ovarian cycle. Our mathematical model suggests that the level of excitability within the ARC kisspeptin network—the propensity of kisspeptin neurons to signal and activate each other—is one of the key parameters modulated in different stages of the cycle by gonadal steroids. Previous studies have shown that ARC kisspeptin neurons synapse on each other (*Yip et al., 2015*; *Qiu et al., 2016*) and are glutamatergic (*Cravo et al., 2011*; *Qiu et al., 2011*; *Kelly et al., 2013*; *Nestor et al., 2016*; *Qiu et al., 2016*; *Wang et al., 2018*). Based on these findings, we hypothesise that population excitability should be enabled primarily via glutamate signalling. We test our predictions in vivo and show that glutamatergic transmission is an important factor for the pulsatile behaviour of the KNDy network.

## Results

### The dynamic response of the KNDy network to sustained, low-frequency optic stimulation is estrous cycle dependent

Using optogenetics we perturbed the KNDy network to test whether and how sex steroids modulate the system's dynamical response. ARC kisspeptin-expressing neurons were transduced with a Cre-dependent adeno-associated virus (AAV9-EF1-dflox-hChR2-(H134R)-mCherryWPRE-hGH) to express ChR2 (*Figure 1*; see Materials and methods) and were optogenetically stimulated at the estrous and the diestrous phase of the cycle, measuring LH pulse frequency as a readout. Sustained, low-frequency optic stimulation was used to emulate elevated basal activity in ARC kisspeptin neurons or persistent stimulatory signals to the KNDy population from other neuronal populations.

In estrous mice, we find that sustained optogenetic stimulation of ARC kisspeptin neurons at 5 Hz immediately triggers robust LH pulses at a frequency of 2.10 ± 0.24 pulses/hour (*Figure 2A and C & E*), which is in agreement with our previous findings (*Voliotis et al., 2019*) and highlights how pulsatile dynamics can emerge as a population phenomenon without the need of a pulsatile activation signal

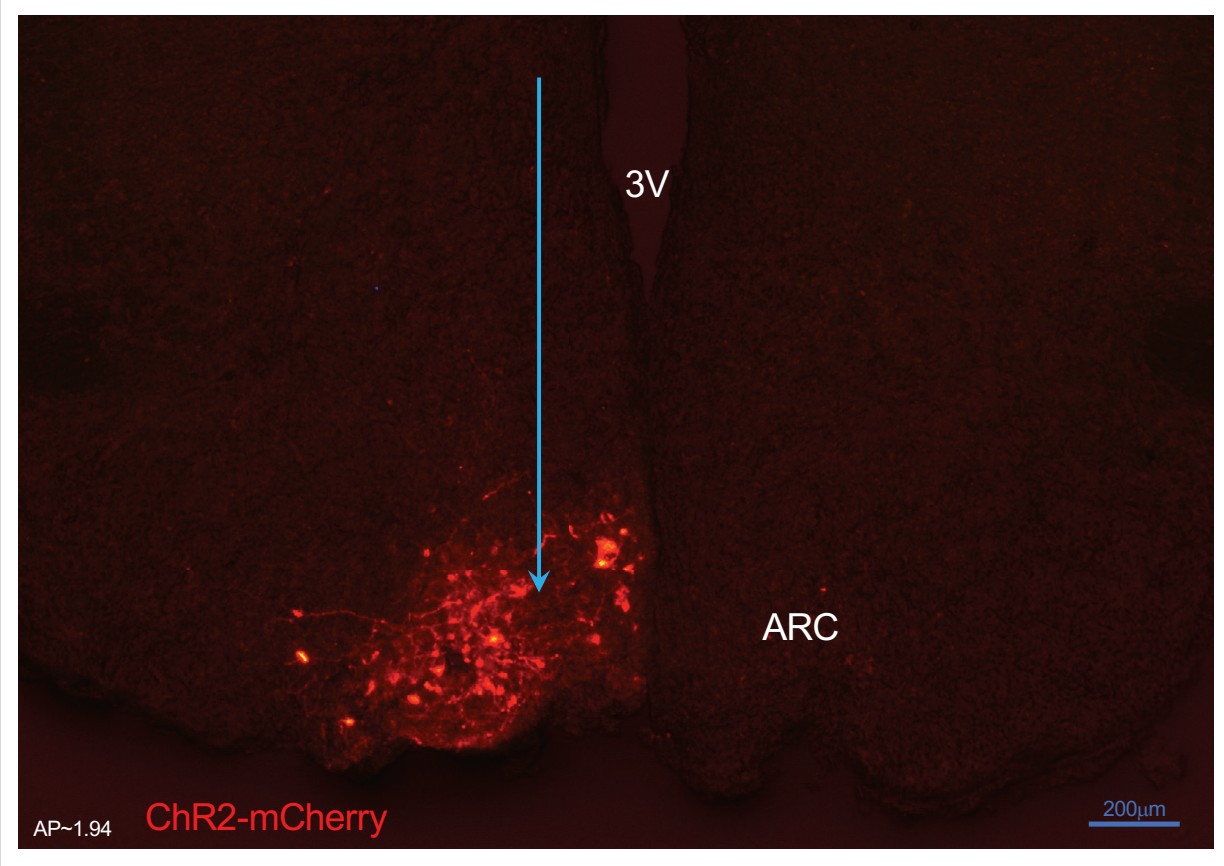

**Figure 1.** Expression of arcuate nucleus (ARC) kisspeptin neurones with ChR2-mCherry in Kiss-Cre mouse. Coronal section showing red mCherry fluorescence positive neurons in the ARC which indicates ChR2 receptor expressing kisspeptin neurones, following unilateral injection of AAV9.EF1. dflox.hChR2(H134R)-mCherry.WPRE.hGH into the ARC of Kiss-Cre mouse. Note the absence of mCherry fluorescence in the other side of ARC. 3 V, Third ventricle.

(*Strogatz, 2018*). In diestrous mice, on the other hand, optogenetic stimulation of ARC kisspeptin neurons at 5 Hz has a subtle slowdown effect on LH pulse frequency over the 1.5 hr stimulation period (*Figure 2—figure supplement 1*). To investigate this response in greater detail, we revised our experimental protocol, removing the control period and extending the stimulation period to 2.5 hr. With the extended protocol we measure 0.64 ± 0.09 and 0.40 ± 0.13 LH pulses/hr under sustained optic stimulation at 5 and 15 Hz, respectively; these frequencies are significantly lower than the LH pulse frequency we observe in control animals, which receive no optic stimulation (*Figure 2B, D, F*). We note that we observe normal LH pulse frequencies in WT animals receiving sustained optic stimulation for 2.5 hr (*Figure 2—figure supplement 2*).

Our data illustrate how natural variation of ovarian steroids across the ovarian cycle leads to qualitative changes in the dynamical response of ARC kisspeptin neurons to optical stimulation. These changes are most probably driven by the effect that gonadal steroids have on the intrinsic electrophysiological properties of ARC kisspeptin neurons (*Ruka et al., 2016*) and the neuromodulator signalling capacity within the KNDy network (*Vanacker et al., 2017*).

## A mathematical model predicts key mechanisms modulating the behaviour of the KNDy pulse generator across the estrous cycle

Interrogating the KNDy network at different stages of the estous cycle via optic stimulation and measuring the effect on LH pulse frequency allows use of our mathematical model (*Voliotis et al., 2019*) to understand how key system parameters change under gonadal steroid control. The model describes the dynamical behaviour of ARC kisspeptin neurons using three dynamical variables: representing the levels of Dyn, NKB and neuronal activity (*Figure 3A*). Furthermore, rather than focusing

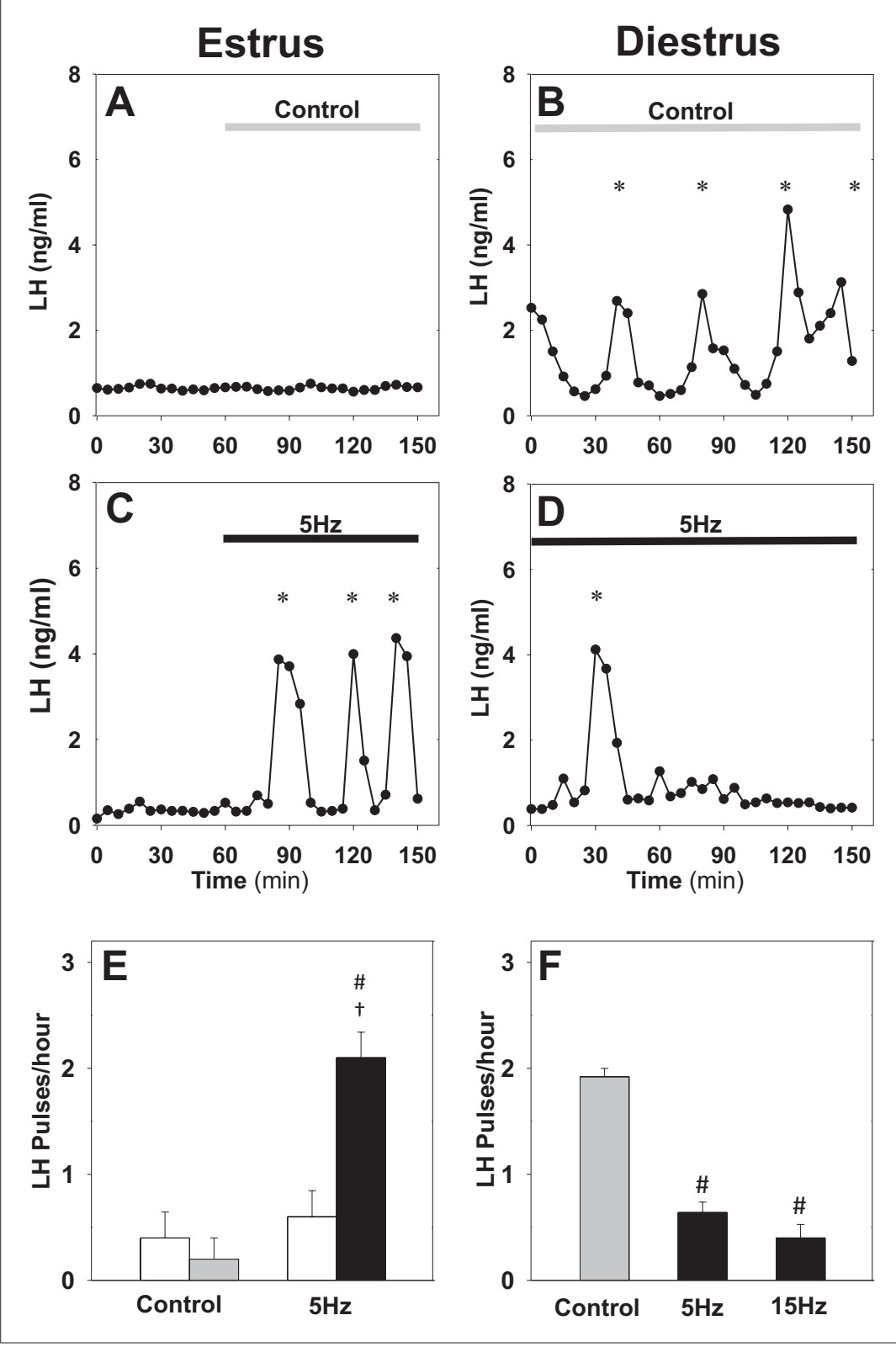

**Figure 2.** Differential effect of optic stimulation of ARC kisspeptin neurons in estrous and diestrous Kiss-Cre mice.
(**A–B**) Representative examples showing LH secretion in response to no stimulation (grey bar) or sustained blue
light (473 nm, 5 ms pulse width, black bar) activation of kisspeptin neurons at 5 Hz in estrous (**C**) and diestrous
(**D**) mice. (**E**) Summary showing mean ± SEM LH pulse frequency over the 60 min control period (white bars) and

*Figure 2 continued on next page*

*Figure 2 continued*

over the subsequent stimulation period (black bar) in estrous mice. (**F**) Summary showing mean ± SEM LH pulse frequency in the control (grey bar) and stimulated (black bars) diestrous mice. *Denote LH pulses. #p < 0.05 vs control; †p < 0.05 vs pre-stimulation; n = 5–6 per group.

The online version of this article includes the following figure supplement(s) for figure 2:

**Figure supplement 1.** Optogenetic stimulation of ARC kisspeptin neurons in diestrous Kiss-Cre mice using the original protocol.

**Figure supplement 2.** Sustained optogenetic stimulation in control WT animals.

on the biophyscal details of regulation, the model postulates that gonadal steroids could potentially modulate the behaviour of the KNDy system across the cycle via acting on four system-level parameters: (i) level of Dyn signalling, (ii) level of NKB signalling, (iii) network excitability (i.e. propensity of neurons in the population to transmit signals to one another), and (iv) basal neuronal activity.

Employing Bayesian inference techniques (see Materials and methods), we sample values for these four parameters, which allow the model to replicate the mean LH frequency we observe experimentally in estrus and diestrus mice with and without 5 Hz optic stimulation (*Figure 2E&F*). Inspection of the dynamical behaviour of the model, using the identified diestrous parameter values, reveals that in response to optic stimulation in diestrus pulsatile dynamics could die out gradually (i.e. there is a transient period before activity shuts down; see *Figure 3B* for an illustrative example), which is confirmed by the delayed inhibition of LH pulses we observed experimentally in diestrous mice.

Next, we focus on how the four key parameters change between diestrus and estrus. We measure the change in each parameter using the log-ratio of its estrous to diestrous value and calculate the covariance matrix of these log-ratios from our set of inferred parameter values. We find a positive (linear) correlation between changes in Dyn and NKB signalling strength, and negative (linear) correlation between changes in NKB signalling strength and network excitability (*Figure 3C*). That is, the model predicts that NKB signalling strength and network excitability are characterised by opposite (in direction) correlations during the transition from diestrus to estrus (one decreasing the other increasing; we note the model predicts that both combinations are possible), whereas NKB and Dyn signalling remain correlated in the same direction (ether increasing or decreasing; we note the model predicts that both combinations are possible). Finally, we apply Principal Component Analysis to study the sensitivity of the system with respect to changes in the four parameters (see Materials and methods). We calculate the principal components in dataset with the inferred parameter changes. Principal components explaining small portions of the variance in the dataset (i.e. principal component with the smallest eigenvalue) correspond to parameter combinations to which the system dynamics are most sensitive (stiff parameter combinations). These combinations are the most critical in terms of regulation as small deviations in how these parameters co-vary result in significant shifts in the system's dynamics. Interestingly, the principal component capturing the smallest share of the variance is comprised of the parameters controlling NKB signalling, Dyn signalling and network excitability in approximately equal portions, and therefore the model predicts that co-ordinated changes in these three parameters should be critical for the observed changes in system dynamics between diestrus and estrus. Interestingly, the second smallest principal component is largely determined by change in the basal activity parameter, suggesting that basal activity is another independent handle for modulating the system's dynamics. Taken together our theoretical findings suggest that co-ordinated changes in KNDy signalling as well as changes in KNDy basal activity may be crucial pathways of regulation across the reproductive cycle.

## Disrupting glutamatergic transmission in the KNDy population blocks lh pulses

Since KNDy neurons are primarily glutamatergic (*Cravo et al., 2011*; *Nestor et al., 2016*; *Qiu et al., 2016*; *Qiu et al., 2018*) and synapse to one another (*Yip et al., 2015*; *Qiu et al., 2016*), we hypothesise that glutamate transmission should directly affect the levels of excitability within the KNDy network. Hence, we disrupt signalling via glutamate receptors to explore in vivo how network excitability affects the ability of the system to generate and sustain LH pulses across the estous cycle.

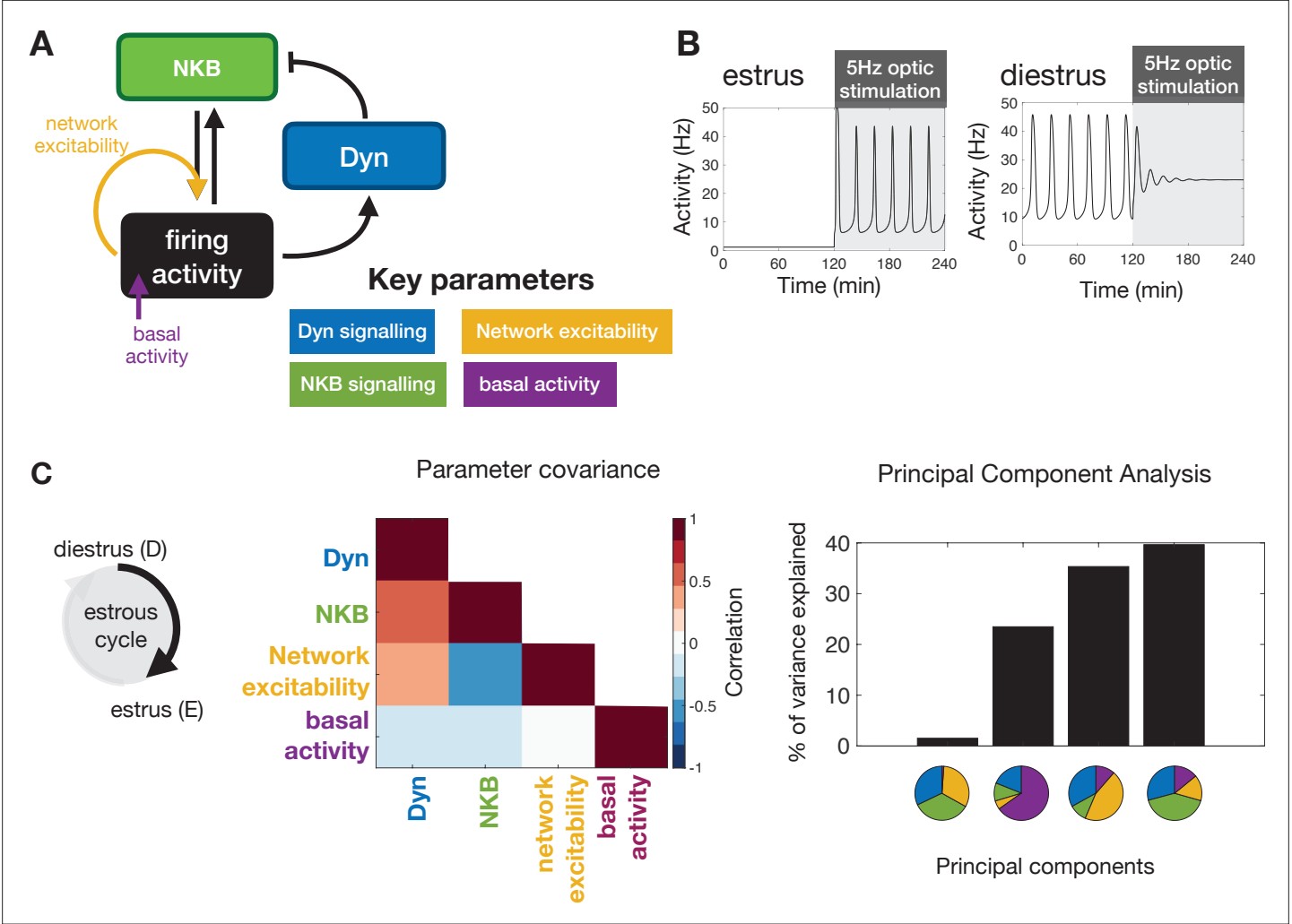

**Figure 3.** Model predictions on the key mechanisms modulating the behaviour of the KNDy pulse generator across the ovarian cycle. (**A**) Schematic illustration of the coarse-grained model of the ARC KNDy population. The model comprises three dynamical variables representing the average levels of Dyn and NKB secreted by the population, and its average firing activity. We hypothesise that four key parameters modulate the behaviour of the system across the ovarian cycle: (**i**) Dyn signalling strength; (**ii**) NKB signalling strength; (**iii**) network excitability; and (**iv**) basal neuronal activity. Estimates for the four parameters in estrus and diestrus are inferred from LH pulse frequency data in estrus and diestrus animals; with or without 5 Hz optic stimulation (**Figure 2E&F**) (**B**) System response to low frequency stimulation during estrus and diestrus, using the maximum a-posteriori estimate of the parameter values inferred from the frequency data. (**C**) Analysis of parameter changes across the cycle. For each of the four parameter $(\theta^i; i = 1, 2, 3, 4)$, the diestrus-to-estrus change is defined as the log-ratio between the corresponding parameter values, that is, $log_{10}\left(\theta^i_{estrus}/\theta^i_{diestrus}\right)$. Normalised covariance (correlation) matrix of parameter changes reveals negative correlation between changes in NKB signalling strength and network excitability, and positive correlation between Dyn signalling strength and both NKB signalling. Eigen-parameters are visualised as pie charts. The eigen-parameter explaining the least of the variance in the posterior distribution corresponds to the stiffest parameter combination to which the system is most sensitive.

The online version of this article includes the following figure supplement(s) for figure 3:

**Figure supplement 1.** Posterior distributions of diestrus-to-estrus parameter changes inferred from data.

First, using Kiss-Cre estrous mice we test whether glutamatergic transmission is necessary for the optogenetic induction of LH pulses. We drive the ARC kisspeptin population using sustained, low-frequency optic stimulation (5 Hz) in the presence of the combined NMDA and AMPA receptor antagonists (AP5 and CNQX, respectively). We find that blocking signalling via glutamate receptors inhibits the capacity of optic stimulation to generate and sustain pulsatile LH secretion (**Figure 4A, B, D**). This is in agreement with the model prediction that network excitability (ability of KNDy neurons to communicate and synchronise) is critical for sustained pulse generation. The combined AP5 and CNQX in the absence of optic stimulation had no effect (**Figure 4C, F**).

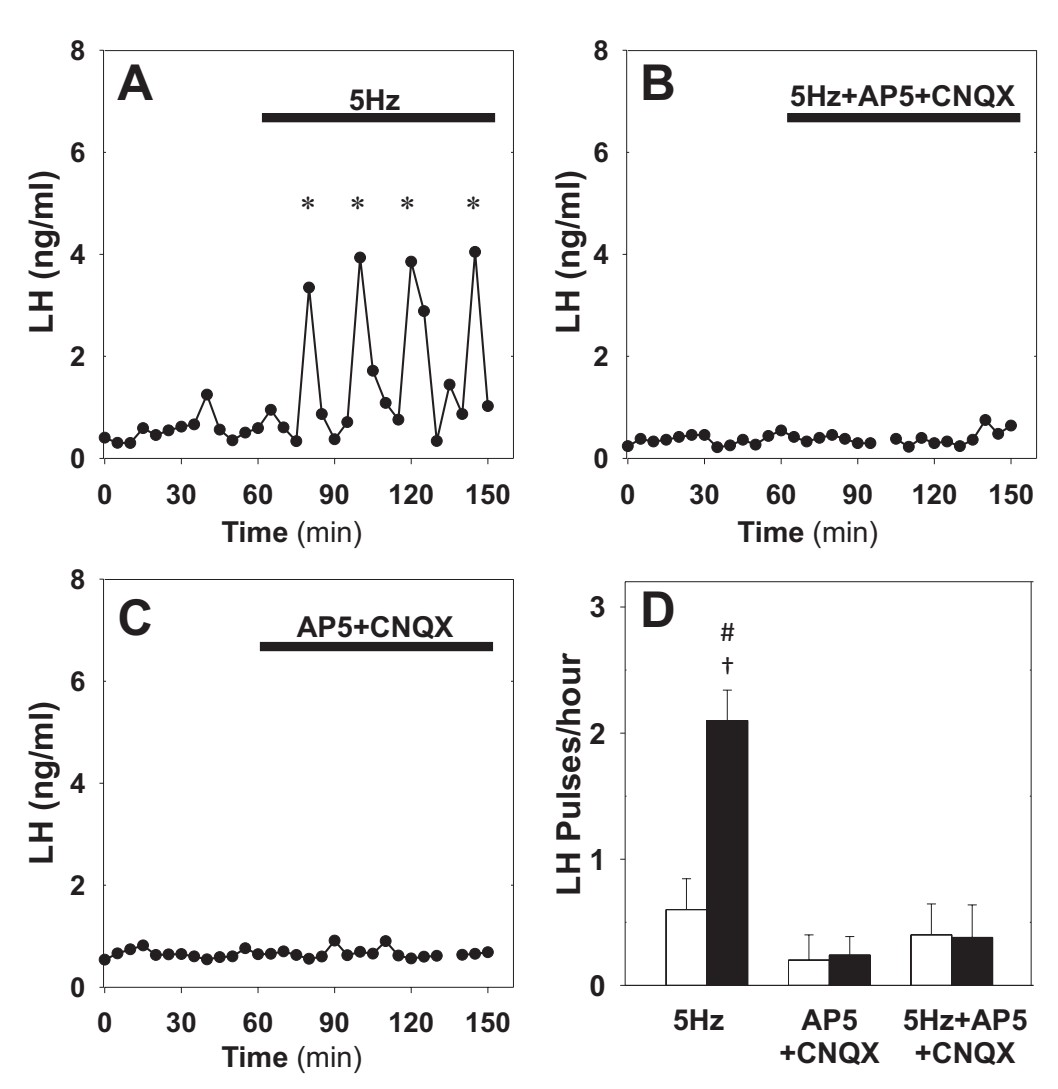

**Figure 4.** Effect of NMDA + AMPA receptor antagonists on pulsatile LH secretion in estrus. Representative examples showing LH secretion in estrous mice in response to optic stimulation (5 Hz blue light, 473 nm, 5 ms pulse width) (**A**) and optic stimulation combined with the NMDA + AMPA receptor antagonist (bolus ICV injection [12 nmol AP5 +5 nmol CNQX] over 5 min, followed by a continuous infusion [20 nmol AP5 and 10 nmol CNQX] for the remaining 90 min) treatment (**B**). NMDA + AMPA receptor antagonist alone had no effect (**C**). (**D**) Summary showing mean ± SEM LH pulse frequency over the 60 min non-stimulatory period (white bars) and over the subsequent 90 min stimulation period or appropriate non-stimulatory period in presence of. NMDA + AMPA receptor antagonist alone (black bar) in diestrous mice. *Denote LH pulses. †$p < 0.05$ vs pre-stimulation. #$p < 0.05$ compared to antagonist treatment groups; n = 5–6 per group.

Next, we test whether glutamatergic transmission is critical for the endogenous LH pulses observed in diestrus. Treatment of diestrous mice with the combined NMDA and AMPA receptor antagonists resulted in a significant reduction of LH pulse frequency from 2.50 ± 0.29–0.45 ± 0.15 pulses/hour (*Figure 5B & D*), confirming that the glutamatergic transmission is indeed critical for sustained pulsatility. Moreover, combining NMDA and AMPA receptor antagonist treatment with low frequency optic stimulation (5 Hz) partially restored LH pulsatility to 1.58 ± 0.17 pulses/hour (*Figure 5C & D*), suggesting low glutamatergic transmission within the KNDy population or from upstream neuronal populations could be offset by other exogenous inputs or elevated basal activity. This finding is in agreement with the model prediction that basal activity and signalling between KNDy neurons are independent pathways of modulating the system's dynamical behaviour.

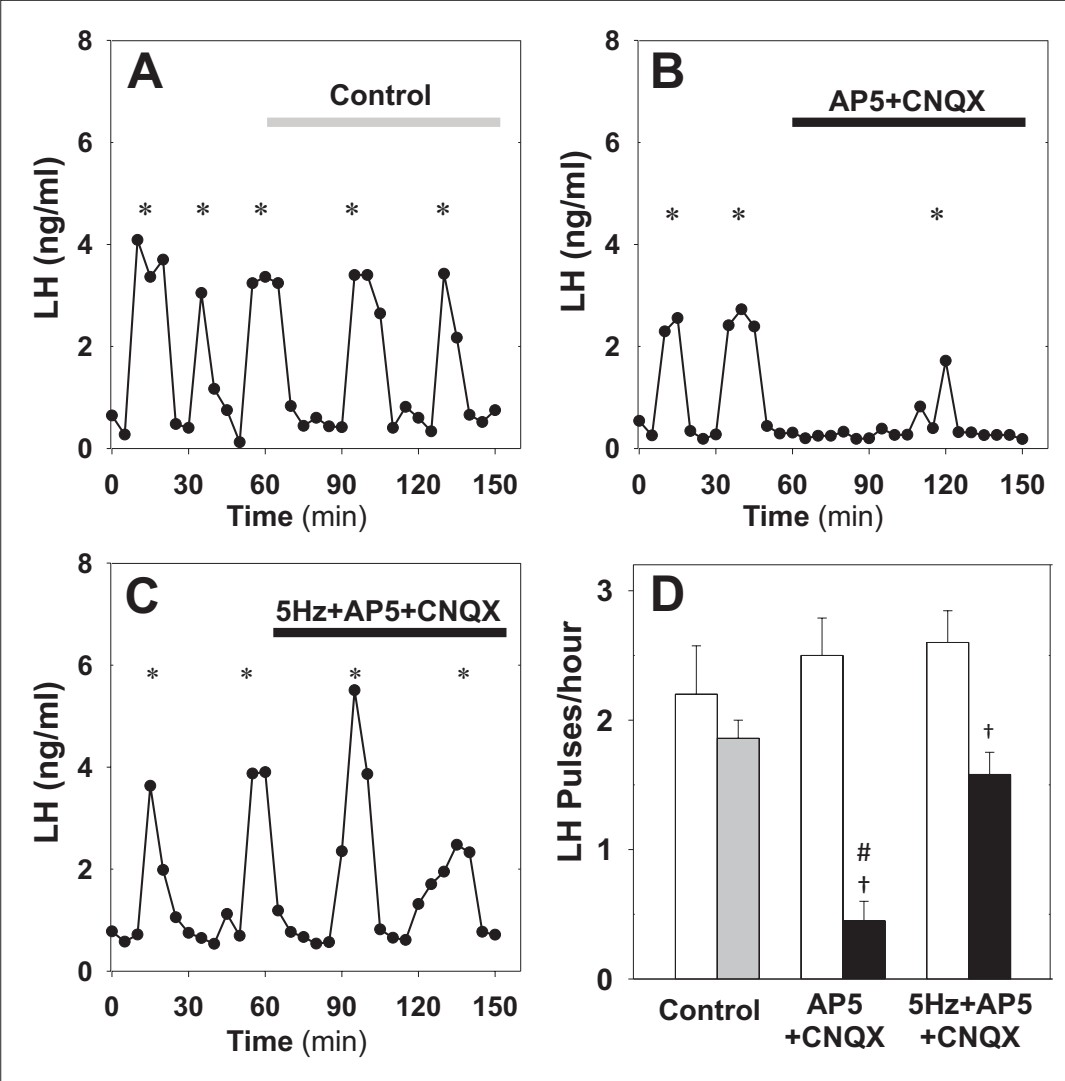

**Figure 5.** Effect of NMDA + AMPA receptor antagonists on pulsatile LH secretion in diestrus. Representative examples showing pulsatile LH secretion in response to ICV administration of aCSF as control (**A**), treatment with NMDA + AMPA receptor antagonists (AP5+ CNQX: bolus ICV injection [12 nmol AP5 +5 nmol CNQX] over 5 min, followed by a continuous infusion [20 nmol AP5 and 10 nmol CNQX] for the remaining 90 min) (**B**) and combined NMDA/AMPA receptor antagonist treatment and sustained optic stimulation (blue light 473 nm, 5 ms pulse width) at 5 Hz (**C**). (**D**) Summary showing mean ± SEM LH pulse frequency over the 60 min non-stimulatory period (white bars) and over the subsequent 90 min stimulation period in control mice (grey bar) and mice receiving treatment (black bar). *Denote LH pulses. [†]p < 0.05 vs pre-stimulation. [#]p < 0.05 compared to 5 Hz stimulation plus antagonist treatment and aCSF control groups; n = 5–6 per group.

## Discussion

Using optogenetics, we perturbed the GnRH pulse generator at different stages of the ovarian cycle aiming to understand how gonadal steroids modulate key properties of the system. Previous studies have shown how the pulsatile activity generated by the kisspeptin neuronal network is modulated across the estrous cycle (**Han et al., 2015**; **McQuillan et al., 2019**). Our data show that the stage of the cycle also has a profound effect on the dynamical response of the kisspeptin population to sustained, low-frequency optic stimulation. Such stimulation triggers acceleration of LH pulses during estrus and deceleration during diestrus. Previously, our mathematical model of the KNDy network has predicted an upper and a lower bifurcation point that determine the system's range of pulsatile behaviour as the system is driven externally (**Voliotis et al., 2019**). Our data suggest that the gonadal state plays a critical role in shifting these bifurcation points by modulating key parameters

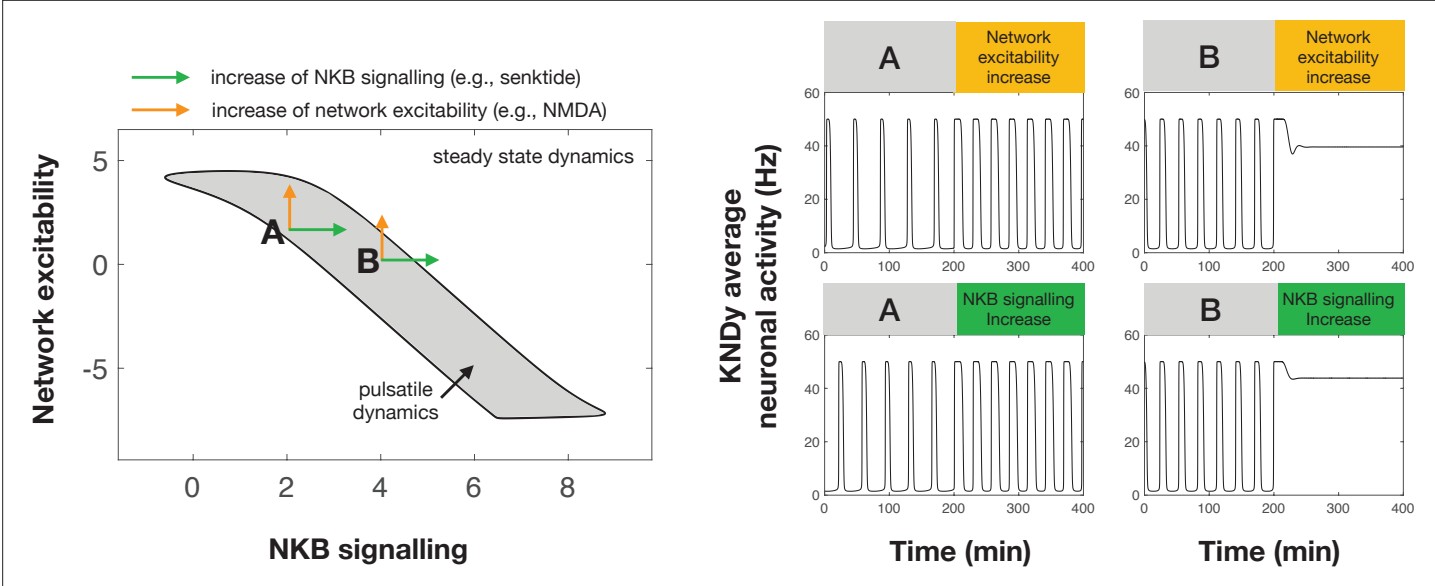

**Figure 6.** Differential effect of perturbations on the dynamics of the pulse generator. Two-parameter (NKB signalling and network excitability) bifurcation diagram showing the region in the parameter space for which the system exhibits pulsatile dynamics (grey area). Two points (denoted by **A** and **B**) illustrate how an increase in NKB signalling or network excitability could have a differential effect on the dynamics of the system. For point A an increase in network excitability or NKB signalling could lead to an increase in the frequency and width of pulses. However, for point B a similar increase leads to pulse inhibition and steady state system dynamics. Furthermore, a negative correlation in how NKB signalling and network excitability co-vary (i.e. aligned with the direction of the pulsatile regime) make the system dynamics less sensitive to small perturbations and enable more robust control over the cyle.

The online version of this article includes the following figure supplement(s) for figure 6:

**Figure supplement 1.** Dynamic behaviour of the KNDY system as a function of dynorphin and NKB singalling.

of the system. In particular, during estrous the system is positioned below the lower bifurcation point and optogenetic stimulation of ARC kisspeptin neurons at 5 Hz moves the system across the lower bifurcation point leading to the sudden emergence of pulsatile behaviour. In contrast, during diestrus the system is within the pulsatile regime and optogenetic stimulation of ARC kisspeptin neurons at frequencies greater than 5 Hz moves the system across the upper bifurcation point and its dynamics relax progressively from pulsatile to quiescent. Our data therefore highlight the critical role of gonadal steroids in modulating the dynamical response of the KNDy network to small changes in basal activity of ARC kisspeptin neurons or in how the population processes external perturbations and afferent inputs.

Using our mathematical model of the system, we gained insight into possible mechanisms via which gonadal steroids modulate the dynamic behaviour of the GnRH pulse generator. Based on the differential effect that optic stimulation had on LH pulse frequency in estrous versus diestrous animals, the model predicted that network excitability is an important parameter, which is actively regulated throughout the ovarian cycle. Importantly, KNDy network excitability is most probably co-regulated with parameters controlling the strength of Dyn and NKB signalling as the system transitions between the different phases of the ovarian cycle. In particular, our analysis predicts (i) a strong negative correlation between changes in NKB signalling strength and changes in network excitability, and (ii) a strong positive correlation in changes between NKB and Dyn signalling. We propose that these regulatory relationships ensure robust control of LH frequency over the estrous cycle (*Figure 6*, *Figure 6—figure supplement 1*). For example, the positive correlation in the regulation of NKB and Dyn signalling enables robust transition between pulsatile and quiescent dynamics, in contrast to negative correlation that would make the system far more sensitive to the magnitude of the change, that is, changes that are too small or too large could fail to trigger LH pulses (see *Figure 6—figure supplement 1*).

Recent transcriptomic data have revealed that treatment of ovariectomised mice with estradiol reduces expression of NKB and Dyn in KNDy neurons, but increases expression of glutamate transporters (vGlut2; leading to increased glutamate neurotransmission and neuronal excitability in the

population) (*Qiu et al., 2018*). These findings are in line with the regulatory relationships predicted by the model and support the hypotheses that (i) correlated changes in the NKB and Dyn signalling strength should be reflected mainly in the expression levels of the two neuropeptides and consequently in their release availability, and (ii) a key mechanism for regulating network excitability could be through the expression of glutamate transporters. In normally cycling animals, these changes should be driven by the combined action of sex steroids. For example, although postovulatory increase in progesterone is linked to deceleration of LH pulses in the luteal-phase, this inhibitory effect of progesterone is conditional on prior exposure to high estradiol levels (*Skinner et al., 1998*) Moreover, data from the rhesus monkey support that estradiol and progesterone could play distinct roles in the deceleration of LH pulses from mid-cycle and throughout the luteal-phase (*O'Byrne et al., 1991*). Our model provides a novel, systems-level understanding of how the genomic changes in the KNDy population link to the dynamic behaviour of the pulse generator. Further experiments will be needed to validate model prediction and link cyclic changes in the sex steroid milieu to genomic pathways dynamically regulating the pulse generator.

The effect of cycle stage on the LH response to sustained optogenetic stimulation is reminiscent of the well documented effect that gonadal steroids have on LH response to various excitatory neurotransmitters and neuropeptides (e.g. NMDA). For instance, investigations in the female monkey revealed an unexplained inhibition of LH in OVX animals following treatment with NMDA, in contrast with the excitatory action of NMDA on LH secretion in the presence of ovarian steroids (*Reyes et al., 1990*; *Reyes et al., 1991*). Similar reversal of action on LH dynamics depending on the underlying ovarian steroid milieu has been documented for various other neurotransmitters and NKB receptor agonists (*Kalra and Kalra, 1983*; *Scorticati et al., 2004*). Our mathematical model supports that ovarian steroids change key parameters of the KNDy network, which control the dynamic behaviour of the system and its response to perturbations. As an illustration, *Figure 6* shows how the dynamic behaviour of the model depends on network excitability and NKB signalling. Since these parameters are governed by gonadal steroids (*Qiu et al., 2018*; *Wang et al., 2018*), it is expected that the underlying steroid milieu will also modulate the effect of perturbations on the dynamics of the system. For instance, the effect of stimulating NKB signalling (e.g. via administration of NK3 receptor agonists) or network excitability (e.g. via NMDA administration) could result in inhibition of the pulse generator if the system is already located closer to the right boundary of the pulsatile dynamics region (e.g. point B in *Figure 6*). Such points correspond to states with high pulse generator activity similar to pulse generator dynamics observed in many animal models after gonadectomy (*Reyes et al., 1990*; *Kinsey-Jones et al., 2012*). In contrast, similar perturbations but from a different point in the parameter space, lying closer to the left edge of the pulsatile region (e.g. point A in *Figure 6*), could result in stimulation of the pulse generator (higher frequency). This illustrative example also highlights that the effect of gonadal steroids on the response of the pulse generator to perturbations is continuous rather than binary, that is, the behaviour of the pulse generator is modulated by the actual continuous levels of gonadal steroids rather than their mere presence of absence. Therefore, the underlying steroid levels could explain seemingly incompatible findings regarding the effect of NKB receptor agonism on LH secretion: ranging from stimulation (*Navarro et al., 2011*) or inhibition (*Sandoval-Guzmán and Rance, 2004*; *Kinsey-Jones et al., 2012*) to no effects (*Navarro et al., 2009*) in rodents.

Based on our model predictions regarding the importance of network excitability, we set out to uncover the role of this parameter on the dynamic response of the network in vivo. We hypothesised that network excitability should depend, partly at least, on the levels of glutamate signalling as ARC kisspeptin neurons are known to be interconnected (*Yip et al., 2015*; *Qiu et al., 2016*) and communicate via glutamate (*Cravo et al., 2011*; *Qiu et al., 2011*; *Nestor et al., 2016*; *Qiu et al., 2016*). Further evidence that estradiol regulates KNDy neuronal excitability (*Qiu et al., 2018*) and that cycle stage regulates spontaneous glutamatergic activity of KNDy neurons (*Wang and Moenter, 2020*) supports the model prediction that network excitability is a critical network property regulated by gonadal steroids. Therefore, to further test this prediction in vivo, we used glutamate receptor (NMDA and AMPA) antagonists to inhibit excitability in the KNDy network. In diestrus animals blocking glutamate receptors (NMDA and AMPA) inhibited LH pulses that were then rescued via low frequency optogenetic stimulation of kisspeptin neurons. Furthermore, in estrus animals, NMDA and AMPA receptor antagonism inhibited the induction of LH pulses via optic stimulation. These experimental findings highlight the complex fine-balanced mechanisms underlying pulse generation by the KNDy network.

In particular, limited network excitability within the KNDy population blocks LH pulsatility but this can be mitigated by elevated basal neuronal activity. Similarly, increased basal neuronal activity can induce pulse generation but this effect can be negated by decreased excitability within the neuronal population. There is a caveat, however, as the glutamate receptor antagonists were given by intra-cerebroventricular injection, and this raises the possibility of having interfered with additional gluta-matergic transmission from afferent populations. To the best of our knowledge, there is no evidence of such afferent populations in rodents, although supporting evidence can be found the sheep litera-ture (*Merkley et al., 2015*). Nevertheless, the possibility of having blocked exogenous glutamatergic inputs does not invalidate our findings, but further supports a key model prediction that basal activa-tion of KNDy neurons (either intrinsic or exogenous) is a critical pathway for modulating the dynamics of the pulse generator. Overall, our model predicts that pulse generation is an emergent property of the KNDy network depending both on single neuron properties such as basal activity and exogeneous activation but also on how the neurons signal and communicate with each other. Our results support this idea, highlighting the critical role of inter-neuronal communication in enabling the population to pulse in synchrony.

## Materials and methods

### Animals

Adult Kiss-Cre heterozygous transgenic female mice aged between 8 and 14 weeks, 25–30 g, were used for experiments (*Yeo et al., 2016*). Breeding pairs were obtained from the Department of Phys-iology, Development and Neuroscience, University of Cambridge, UK and mated in house at King's College London. Genotyping was performed using a multiplex PCR protocol for detection of hetero-zygosity for the Kiss-Cre or wild-type allele as previously described (*Lass et al., 2020*). Only mice with normal estrous cycles were used. Daily vaginal smears were performed for the detection of the estrous and diestrous stages of the ovarian cycle. Mice were singularly housed and provided with food (stan-dard maintenance diet; Special Dietary Services, Wittam, UK) and water ad libitum while being kept under a 12:12 hr light/dark cycle (lights on 0700 h) at 23°C ± 2°C. All animal procedures performed were approved by the Animal Welfare and Ethical Review Body Committee at King's College London and conducted in accordance with the UK Home Office Regulations.

### Surgical procedures

Stereotaxic injection of AAV9-EF1-dflox-hChR2-(H134R)-mCherry-WPRE-hGH (4.35 × $10^{13}$ GC/ml; Penn Vector Core; University of Pennsylvania, PA, USA) for targeted expression of channelrhodopsin (ChR2) in ARC kisspeptin neurons was done under aseptic conditions. The mice were anaesthetised using ketamine (Vetalar, 100 mg/kg, i.p.; Pfizer, New York City, NY, USA) and xylazine (Rompun, 10 mg/kg, i.p.; Bayer, Leverkusen, Germany). Kiss-Cre female mice (n = 12) or wilt-type (n = 3) were secured in a Kopf Instruments motorised stereotaxic frame (Kopf Instruments, Tujunga, CA, USA) and surgical procedures on the brain were performed using a Robot Stereotaxy system (Neurostar, Tubingen, Germany). Stereotaxic injection coordinates used to target the ARC were obtained from the mouse brain atlas of *Paxinos and Franklin, 2004* (0.25 mm lateral, 1.94 mm posterior to bregma and at a depth of 5.8 mm). A skin incision was made and a small hole was drilled in the skull above the location of the ARC. A 2 µl Hamilton micro-syringe (Esslab, Essex, UK) was attached to the robot stereotaxy and used to inject 0.3 µl of the AAV-construct into the ARC, unilaterally, at a rate of 100 nl/ min. After the injection, the needle was left in position for 5 min and then slowly lifted over 1 min. The same coordinates as the injection site were then used to insert a fibre-optic cannula (200 µm, 0.39 NA, 1.25 mm ceramic ferrule; Thorlabs, LTD, Ely, UK); however, a depth of 5.78 mm was reached to ensure the fibre-optic cannula was situated immediately above the injection site. Additionally, an intra-cerebroventricular (ICV) fluid guide cannulae (26 gauge; Plastics One) targeting the lateral ventricle (coordinates: 1.1 mm lateral, 1.0 mm posterior to bregma and at a depth of 3.0 mm) was chronically implanted. Dental cement (Superbond C&B kit Prestige Dental Products, Bradford UK) was used to fix the cannulae in place and the skin incision was sutured. A one week recovery period was given post-surgery. After this period, the mice were handled daily to acclimatise them to the tail-tip blood sampling procedure (*Steyn et al., 2013*). Mice were left for 4 weeks to achieve effective opsin expres-sion before experimentation.

## Validation of AAV injection site and fibre optic and ICV cannula position

Once experiments were completed, mice were given a lethal dose of ketamine and transcardially perfused for 5 min with heparinised saline, followed by 10 min of ice-cold 4 % paraformaldehyde (PFA) in phosphate buffer, pH 7.4, for 15 min using a pump (Minipuls; Gilson). Brains were collected immediately and post fixed at 4 °C in 15 % sucrose in 4 % PFA and left to sink. They were then transferred to 30 % sucrose in PBS until they sank. The brains were then snap-frozen on dry ice and stored at –80 °C. Using a cryostat, every third coronal brain section (30 µm) was collected between –1.34 mm and –2.70 mm from bregma and sections were mounted on microscope slides, left to air-dry and cover slipped with ProLong Antifade mounting medium (Molecular Probes, Inc, OR, USA). Verification and evaluation of the injection site was performed using an Axioskop 2 Plus microscope equipped with axiovision 4.7 (Zeiss). One of 12 Kiss-Cre mice failed to show mCherry fluorescence in the ARC and was excluded from the analysis.

## Experimental design and blood sampling for LH measurement

For measurement of LH pulsatility during optogenetic stimulation, the tip of the mouse's tail was removed with a sterile scalpel for tail-tip blood sampling (*Czieselsky et al., 2016*). The chronically implanted fibre-optic cannula was attached to a multimode fibre-optic rotary joint patch cables (Thorlabs) via a ceramic mating sleeve. This allows for freedom of movement and blue light delivery (473 nm wavelength) using a Grass SD9B stimulator controlled DPSS laser (Laserglow Technologies) during optogenetic stimulation.

The experimental protocol involved an hour long acclimatisation period, followed by 2.5 hr of blood sampling, where 5 µl of blood was collected every 5 min. For estrous and diestrous mice, optic stimulation was initiated after 1 hr of control blood sampling and was sustained for 1.5 hr. Optic stimulation was delivered as 5 ms pulses of light at 5 Hz with the laser intensity measured at the tip of the fibre-optic patch cable set to 5 mW (*Voliotis et al., 2019*). Additionally, in separate experiments, diestrous mice were optically stimulated at five or 15 Hz for 2.5 hr, that is entire blood sampling period. Control mice (in estrus or diestrus) received no optic stimulation. Wild-type mice (estrus and disetrus) received 5 Hz optic stimulation to verify that our optic stimulation protocol had no undesirable effects on LH secretion.

Neuropharmacological manipulation of glutamatergic signalling was performed using a combination of NMDA (AP5, Tocris, Abingdon, UK) and AMPA (CNQX, Alpha Aesar, Heysham, UK) receptor antagonist treatment with or without simultaneous optogenetic stimulation. The animals were prepared for optogenetic experimentation as described above with additional preparation of the ICV injection cannula. Immediately after connection of the fire-optic cannula, the ICV injection cannula with extension tubing, preloaded with drug solution (AP5 and CNQX dissolved in artificial CSF) or artificial CSF alone as control, was inserted into the guide cannula. The extension tubing, reaching outside of the cage, was connected to a 10 µl Hamilton syringe mounted in an automated pump (Harvard Apparatus) to allow for remote micro-infusion without disturbing the animals during experimentation. After a 55 min control blood sampling period, as described above, and 5 min before the onset of optic stimulation, a bolus ICV injection of drug solution (12 nmol AP5 and 5 nmol CNQX in 2.3 µl) was given over 5 min, followed by a continuous infusion (20 nmol AP5 and 10 nmol CNQX in 5.6 µl) for the remaining 90 min of experimentation. Artificial CSF controls, with or without optic stimulation, received the same ICV fluid regime. When no optic stimulation was applied the same ICV administration and blood sampling regimen described was applied. Stimulation and non-stimulation protocols were implemented in random order for Kiss-Cre mice.

The blood samples were snap-frozen on dry ice and stored at –80 °C until processed. In-house LH enzyme-linked immunosorbent assay (LH ELISA) similar to that described by Steyn et al. was used for processing of the mouse blood samples (*Steyn et al., 2013*). The mouse LH standard (AFP- 5,306 A; NIDDK-NHPP) was purchased from Harbor-UCLA along with the primary antibody (polyclonal antibody, rabbit LH antiserum, AFP240580Rb; NIDDK-NHPP). The secondary antibody (donkey anti-rabbit IgG polyclonal antibody [horseradish peroxidase]; NA934) was from VWR International. Validation of the LH ELISA was done in accordance with the procedure described in *Steyn et al., 2013* derived from protocols defined by the International Union of Pure and Applied Chemistry. Serially diluted mLH standard replicates were used to determine the linear detection range. Nonlinear regression analysis

was performed using serially diluted mLH standards of known concentration to create a standard curve for interpolating the LH concentration in whole blood samples, as described previously (**Voliotis et al., 2019**). The assay sensitivity was 0.031 ng/mL, with intra- and inter-assay coefficients of variation of 4.6% and 10.2%, respectively.

## LH pulse detection and statistical analysis

Dynpeak algorithm was used for the detection of LH pulses (**Vidal et al., 2012**). The differential effect of optogenetic stimulation on LH pulsatility in estrus and diestrus was determined by looking at the frequency of LH pulses. For mice in estrus and for the neuropharmacological experiments, the mean ± SEM of LH pulses per hour were compared between the 60 min pre-stimulation/drug delivery control period and subsequent 90 min stimulation period. For mice in diestrus, the mean ± SEM of LH pulses per hour were compared between controls, 5 Hz and 15 Hz treatment groups, as optic stimulation was applied from the beginning of blood sample period. No optic stimulation was applied to control animals, however the same time points were compared. The frequency of LH pulses in the 90 min optic stimulation/drug delivery period was also compared between treatment groups. Mann-Whitney Rank Sum test was used to access LH frequency differences between groups and determine statistical significance (p < 0.05). LH data publicly available from http://doi.org/doi:10.18742/RDM01-750.

## Mathematical model of the KNDy network

We used a modified version of our previously published mathematical model of the KNDy network (**Voliotis et al., 2019**). The model offers a high-level overview of the system, wrapping many biophysical details into a coarse grain description for the sake of simplicity and brevity. Importantly such a parsimonious model fits best to the high-level, holistic in vivo approach we use to study the system. Briefly, the model describes the ARC kisspeptin population in terms of three variables: $D$, the average concentration of Dyn secreted by the population; $N$, the average concentration of NKB secreted by the population; and $v$, the average firing activity of the population, measured in spikes/min. The variables obey the following set of coupled ordinary differential equations (ODEs):

$$\frac{dD}{dt} = f_D(v) - d_D D; \tag{1}$$

$$\frac{dN}{dt} = f_N(v, D) - d_N N; \tag{2}$$

$$\frac{dv}{dt} = f_v(v, N) - d_v v. \tag{3}$$

Parameters $d_D$, $d_N$ and $d_v$ control the characteristic timescale of each variable. The model describes Dyn and NKB secretion as independent processes based on the observation that Dyn and NKB are packaged in separate vesicles (**Murakawa et al., 2016**). The secretion rates of the two neuropeptides are given by:

$$f_D(v) = k_D \frac{v^2}{v^2 + K_v^2};$$

$$f_N(v, D) = k_N \frac{v^2}{v^2 + K_v^2} \frac{K_D^2}{D^2 + K_D^2}.$$

In the equations above neuronal activity ($v$) stimulates secretion of both neuropeptides, and Dyn represses NKB secretion. The maximum secretion rate for the two neuropeptides is controlled by parameters $k_D$ and $k_N$ and we refer to these parameters as the strength of Dyn and NKB singalling respectively. Furthermore, we assume that distinct modes of Dyn and NKB regulation (e.g. in terms of their synthesis and depletion rate, intracellular transport, packaging dynamics) are reflected in their secretion rate and therefore model Dyn and NKB regulation throughout the estrous cycle as changes of parameters $k_D$ and $k_N$. The effector levels at which saturation occurs are controlled via parameters $K_v$ and $K_D$. Here, we are interested in investigating the effect of network excitability on the dynamics therefore we modify the equation for the neuronal activity, $v$, by setting:

$$f_v(v, N) = v_0 \frac{1 - exp(-I)}{1 + exp(-I)}; I = -log \frac{1 - b}{1 + b} + k_v \left( e + \frac{N^2}{N^2 + K_N^2} \right) v.$$

Here, we have introduced parameter $k_v$ capturing the intrinsic network excitability, that relates to the strength of the synaptic connections between KNDy neurons that are essential for synchronising

**Table 1.** Model parameters values.

| No | Parameter | Description | Value | Ref. |
|----|-----------|-------------|-------|------|
| 1 | $d_D$ | Dyn degradation rate | 0.25 min$^{-1}$ | *Voliotis et al., 2019* |
| 2 | $d_N$ | NKB degradation rate | 0.25 min$^{-1}$ | *Voliotis et al., 2019* |
| 3 | $d_v$ | Firing rate reset rate | 10 min$^{-1}$ | *Qiu et al., 2016* |
| 4 | $k_D$ | Dyn singalling strength | inferred | |
| 5 | $k_N$ | NKB signalling strength | inferred | *Ruka et al., 2016* |
| 6 | $k_v$ | Network excitability | inferred | |
| 7 | $v_0$ | Maximum rate of neuronal activity increase | 30000 spikes min$^{-2}$ | *Qiu et al., 2016* |
| 8 | $K_D$ | Dyn IC$_{50}$ | 0.3 nM | *Yasuda et al., 1993* |
| 9 | $K_N$ | NKB EC$_{50}$ | 32 nM | *Seabrook et al., 1995* |
| 10 | $K_v$ | Firing rate for half-maximal NKB and Dyn secretion | 1200 spikes min$^{-1}$ | *Dutton and Dyball, 1979* |
| 11 | $b$ | Basal activity | inferred | |
| 12 | e | NKB-independent contribution to network excitability | inferred | |

neural activity and enabling pulse generation. We note that this parameter will be directly affected by any neuromodulator (including Glutamate) that affects KNDy activity as well as by processes that affect KNDy neurons' synaptic density. Furthermore, parameter $b$ controls the basal neuronal activation of the population, which could stem from synaptic noise or afferent inputs (extrinsic to the network). We assume that both $k_v$ and $b$ are regulated throughout the estrous cycle. Finally, $v_0$ is the maximum rate at which the firing rate increases in response to synaptic inputs $I$. Note the stimulatory effect of NKB (which is secreted at the presynaptic terminal) on neuronal activity (*Qiu et al., 2016*). The full list of model parameters is given in *Table 1*.

## Parameter inference

We used Approximate Bayesian Computation (ABC) based on sequential Monte Carlo (SMC) (*Toni et al., 2009*) to infer four key model parameters (Dyn signaling strength, $k_D$ ; NKB signalling strength, $k_N$ ; network excitability $k_v$ ; and basal activity, $b$) in the estrous and diestrous phase of the ovarian cycle. For inference we used the average LH inter-pulse interval observed in four different settings: estrous animals without optic stimulation ($I_E$) and with 5 Hz optic stimulation ($I_{E+5Hz}$); diestrus animals without optic stimulation ($I_D$) and with 5 Hz optic stimulation ($I_{D+5Hz}$). Model simulations were generated in Matlab using function ode45 under the four different settings for 6000 min and by calculating the frequencies after discarding the initial 1000 min. The following discrepancy function was used to compare simulated, $D = \left(I_E^*, I_{E+5Hz}^*, I_D^*, I_{D+5Hz}^*\right)$ , and experimental, $D = \left(I_E, I_{E+5Hz}, I_D, I_{D+5z}\right)$ , data:

$$d\left(D, D^*\right) = \sum_{i=1}^{4} \left|D_i - D_i^*\right|$$

Furthermore, for the ABC SMC algorithm the size of the particle population was set to 500 and the algorithm was run for $T = 4$ populations with corresponding tolerance levels $\varepsilon_i = 54 - 2i, i = 0, \ldots, 26$. Log-uniform prior distributions were used to explore the behaviour of the model under wide parameter ranges: $\log_{10}\left(k_D\right) \sim Uniform\left(-3,3\right)$ ; $log_{10}\left(k_N\right) \sim Uniform\left(-3,3\right)$ ; $log_{10}\left(k_v\right) \sim Uniform\left(-3,3\right)$ ; and $\log_{10}\left(b\right) \sim Uniform\left(-3,0\right)$. All remaining parameters were fixed to values found in the literature (see *Table 1*). For each parameter, an independent $\log_{10}$-normal perturbation kernel with variance 0.05 was used. Matlab code can be found at https://git.exeter.ac.uk/mv286/kndy-parameter-inference (*Voliotis, 2021*, copy archived at swh:1:rev:f5585f5f09d62a2569d21def31dc779774bc2476)

## Sensitivity and principal component analysis

We used principal component analysis to study the sensitivity of the system with respect to changes in the four inferred parameters (*Toni et al., 2009*). We calculate the principal components in the dataset (sampled posterior distribution) of the inferred parameter changes. Principal component analysis produces a set of linearly uncorrelated eigen-parameters explaining the variance of the inferred changes (in the sampled posterior distribution.) These eigen-parameters are linear weighted combinations of the initial parameters. The eigen-parameter explaining the least of the variance in the posterior distribution corresponds to the stiffest parameter combination. That is small deviations from the inferred way these parameters co-vary would lead to changes in the model behaviour that make it incompatible with the data.

## Acknowledgements

The authors gratefully acknowledge the financial support of the EPSRC via grant EP/N014391/1 (KTA and MV), and BBSRC via grants BB/S000550/1 and BB/S001255/1 (KTA, KOB, MV, XFL).

## Additional information

### Funding

| Funder | Grant reference number | Author |
|---|---|---|
| Engineering and Physical Sciences Research Council | EP/N014391/1 | Margaritis Voliotis Krasimira Tsaneva-Atanasova |
| Biotechnology and Biological Sciences Research Council | BB/S000550/1 | Margaritis Voliotis Xiao Feng Li Kevin O'Byrne Krasimira Tsaneva-Atanasova |
| Biotechnology and Biological Sciences Research Council | BB/S001255/1 | Margaritis Voliotis Xiao Feng Li Kevin O'Byrne Krasimira Tsaneva-Atanasova |

The funders had no role in study design, data collection and interpretation, or the decision to submit the work for publication.

### Author contributions

Margaritis Voliotis, Conceptualization, Investigation, Methodology, Software, Writing – original draft; Xiao Feng Li, Conceptualization, Data curation, Investigation, Writing – review and editing; Ross Alexander De Burgh, Deyana Ivanova, Caitlin McIntyre, Data curation, Investigation; Geffen Lass, Investigation; Kevin O'Byrne, Conceptualization, Investigation, Writing – original draft; Krasimira Tsaneva-Atanasova, Conceptualization, Investigation, Writing – original draft, Writing – review and editing

### Author ORCIDs

Margaritis Voliotis (iD) http://orcid.org/0000-0001-6488-7198
Krasimira Tsaneva-Atanasova (iD) http://orcid.org/0000-0002-6294-7051

### Ethics

All animal procedures performed were approved by the Animal Welfare and Ethical Review Body Committee at King's College London (PP4006193 ) and conducted in accordance with the UK Home Office Regulations.

### Decision letter and Author response

Decision letter https://doi.org/10.7554/eLife.71252.sa1
Author response https://doi.org/10.7554/eLife.71252.sa2

## Additional files

### Supplementary files
• Transparent reporting form

### Data availability
The data and the code are publicly available via the following open access repositories: http://doi.org/doi:10.18742/RDM01-750   https://git.exeter.ac.uk/mv286/kndy-parameter-inference.git (copy archived at https://archive.softwareheritage.org/swh:1:rev:f5585f5f09d62a2569d21def31dc779774bc2476).

The following dataset was generated:

| Author(s) | Year | Dataset title | Dataset URL | Database and Identifier |
|---|---|---|---|---|
| de Burgh R | 2021 | Modulation of pulsatile GnRH dynamics along the reproductive cycle - the role of excitability within the arcuate kisspeptin network | https://doi.org/10.18742/RDM01-750 | figshare, 10.18742/RDM01-750 |

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
