## [Editor Report]

Successful reproduction requires luteinizing hormone that is secreted from the pituitary gland in pulses which vary over the female reproductive cycle. The pulses arise by patterned secretion of a hypothalamic 'releasing factor', the secretion of which is itself governed by a population of hypothalamic neurons that express the neuropeptide kisspeptin. The paper by Voliotis et al. combines novel experimental evidence from transgenic mice with an elegant mathematical model to analyze how the kisspeptin neurons generate the varying pulsatile patterns. Testing this model in a way that will either confirm or refute it, may prove challenging.

---

## [Decision Letter]

**Decision letter after peer review:**

Thank you for submitting your article "Modulation of pulsatile GnRH dynamics across the ovarian cycle: the role of glutamatergic transmission in the arcuate kisspeptin network" for consideration by *eLife*. Your article has been reviewed by 2 peer reviewers, and the evaluation has been overseen by a Reviewing Editor and Catherine Dulac as the Senior Editor. The following individuals involved in review of your submission have agreed to reveal their identity: Robert Goodman (Reviewer #1); Gareth Leng (Reviewer #2).

Essential revisions:

1) Given the model is based on somewhat loosely define "neuronal excitability" there is a question as to whether the model is truly testable.

2) Much of your findings are based on the role of glutamate and yet the effects of glutamate do not appear to be included in the model

3) Please clarify whether the effects of NKB and dynorphin in this study are consistent with or in conflict with versus previous publications by yourself and others..

*Reviewer #1 (Recommendations for the authors):*

The original model (J Neurosci 39: 9738, 2019) described by this group incorporated dynorphin as a negative feedback signal onto NKB, a characteristic that is no longer depicted in the model proposed in Figure 3 of this manuscript. Has the model changed in this regard?

L178-79: Inclusion of the minimal detectable concentration of LH would be helpful

L 189-192: Since pulse frequency is not a continuous variable, non-parametric statistics are more appropriate

L 276-281 and 316-320: Although one should always be careful about interpreting individual pulse patterns, the response to 5Hz stimulation on diestrus appears to be more robust in Figure 2 that in Supplemental Figure 2. Perhaps more importantly, the pulse patterns in two of the three mice in the supplemental figure are not consistent with pattern produced by the model (Figure 3B).

L 503-504: There are data in sheep for glutamatergic inputs to KNDy neurons from non-KNDy afferents that are as numerous (luteal phase) or more numerous (LH surge) than glutamatergic input from KNDy neurons (Merkley et al., J Neuroendo 27: 624, 2015).

*Reviewer #2 (Recommendations for the authors):*

It is hard to see how this role of glutamate might be incorporated in this style of model. In the model auto-excitation is modelled by the excitatory action of neurokinin, so perhaps another variable to represent activity-dependent glutamate signaling might be added with very different dynamics, but I don't know that it is necessarily worth complicating the model in quite that way. Perhaps this is something that merits careful discussion.

[Editors' note: further revisions were suggested prior to acceptance, as described below.]

Thank you for submitting your revised article "Modulation of pulsatile GnRH dynamics across the ovarian cycle via changes in the network excitability and basal activity of the arcuate kisspeptin network" for consideration by *eLife*. Your article has been reviewed by 2 peer reviewers, and the evaluation has been overseen by a Reviewing Editor and Catherine Dulac as the Senior Editor. The following individuals involved in review of your submission have agreed to reveal their identity: Robert Goodman (Reviewer #1); Gareth Leng (Reviewer #2).

Essential revisions:

One of the reviewers has asked for a few additional points for clarification and some speculation by you on the mechanism of altered neurotransmission which might aid in formulating tests of your model.

1) The model is explained much more clearly – but there are a few further simplifications that might be made. For example n1,n2, n3 and n4 are all just Hill coefficients and are all = 2; so what seem to be four parameters are just a single constant. Kv,1=Kv,2 – replace with a single parameter? It's just a bit harder than it need be to work out exactly what the model structure is.

2) The manuscript would be improved if the authors were to say whether they think that what they call changes in signaling strength reflect changes in vesicle content, vesicle availability for release, mechanisms coupling activity to vesicle release, receptor availability or post-receptor signaling. I am left uncertain what experiments might be proposed to really test this model (ie potentially refute it), but if any of these were specified then designing a critical test would be simpler.

---

## [Author Response]

Essential revisions:1) Given the model is based on somewhat loosely define "neuronal excitability" there is a question as to whether the model is truly testable.

A key strength of our mathematical model is that it is simple yet able to explain complex behaviour. To achieve this the model makes certain assumptions, which in the revised version of the manuscript are made explicit (see ‘Mathematical model of the KNDy network’ section). For instance, an important point raised by reviewer #1 is that “any pharmacological manipulation that affects the level of excitability of the network might have similar effects as the blockade of glutamatergic transmission reported here”. This is true as the model is agnostic regarding what drives excitability in the network other than it should be enabled via KNDy-to-KNDy communication. This allows us to generalise our results obtained using glutamate antagonists to provide a putative explanation regarding the apparent paradoxical effects that other neurotransmitters have on LH pulses. Therefore, the model fully serves its purpose of providing insight into this complex system and what changes over the course of the estrous cycle, while more complex models (incorporating specific biophysical mechanisms) along with more advanced experimental techniques (GRIN lens) will be needed to discriminate the major drivers of intra-KNDy communication and dissect these from external modulators.

2) Much of your findings are based on the role of glutamate and yet the effects of glutamate do not appear to be included in the model

As we stated above a key characteristic of our model is the simple, top-down description of the KNDy network. Rather than focusing on specific biophysical details the model describes the systems in higher-level (network) terms, for instance, it describes network excitability (the ability of KNDy neurons to communicate and synchronise), instead of describing the intricacies of glutamate neurotransmission between KNDy neurons. That does not mean the model cannot be tested rather it admits that the model needs refinement to tackle more specific questions at the level of individual neurons. Furthermore, a major concern of reviewer #2 is that glutamate neurotransmission has several roles in the system, i.e., glutamate could relay signals from afferent neuronal populations (external inputs) as well between KNDy neurons (enabling them to excite each other and synchronise). These two roles have been incorporated into our model and we now describe how they link to distinct parameters in our model. Accordingly, we have revised our manuscript to discuss how the model provides an simple yet useful representation of the KNDy system, and opens the way for more detailed models adapted to tackle specific questions regarding the regulation of the KNDy network along the reproductive cycle.

3) Please clarify whether the effects of NKB and dynorphin in this study are consistent with or in conflict with versus previous publications by yourself and others..

One of the key finding of fitting the model parameters to our experimental data is that there is a positive correlation between changes in NKB and dynorphin signalling along the estrous cycle. This is consistent with transcriptomic data showing that treatment of ovariectomised mice with estradiol reduced expression of both NKB and dynorphin (Qiu et 2018). As pointed out by reviewer #1 this positive correlation might seem counterintuitive at first since the two neuropeptides have opposite effect on LH secretion. However, the opposite effect on LH secretion is perhaps what underlies the co-regulation of the two neuropeptides for the purpose of robustly modulating LH frequency. A positive correlation in the changes of NKB and dynorphin signalling gives rise to robust control over LH frequency, whereas a negative correlation would make LH pulsatility highly sensitive to the relative expression of NKB and dynorphin and would require additional fine-tuning (see Figure 6—figure supplement 1). This important point has now been included and discussed in the revised manuscript.

Reviewer #1 (Recommendations for the authors):The original model (J Neurosci 39: 9738, 2019) described by this group incorporated dynorphin as a negative feedback signal onto NKB, a characteristic that is no longer depicted in the model proposed in Figure 3 of this manuscript. Has the model changed in this regard?

We thank the reviewer for pointing this out. We have now revised Figure 3 to illustrate that dynorphin exerts a negative feedback onto NKB.

L178-79: Inclusion of the minimal detectable concentration of LH would be helpful

We have now revised the text to include the minimal detectable concentration of LH (L180).

L 189-192: Since pulse frequency is not a continuous variable, non-parametric statistics are more appropriate

We agree with the referee that a non-parametric test is more suitable and therefore we have re-analysed our data using the Mann-Whitney Rank Sum test and updated the methods section (L192-194). We found no changes in the statistical significance of our findings.

L 276-281 and 316-320: Although one should always be careful about interpreting individual pulse patterns, the response to 5Hz stimulation on diestrus appears to be more robust in Figure 2 that in Supplemental Figure 2. Perhaps more importantly, the pulse patterns in two of the three mice in the supplemental figure are not consistent with pattern produced by the model (Figure 3B).

This is true, as we describe in the main text the data illustrated in Figure 2—figure supplement 1 were obtained using a different protocol, i.e., 60-min control period followed by 90 minutes of optic stimulation. As Figure 2—figure supplement 1 shows there is a tendency for LH pulses to slow down over the 90 min stimulation interval (panels A and C), however there wasn’t a clear cut difference and this led us to revise our protocol and stimulate for longer (150min). With the revised protocol we found a significant effect of optic stimulation on diestrous mice (Figure 2). We have now revised the main text (L293-297) and legend of Figure 2—figure supplement 1 to clarify this for the reader.

L 503-504: There are data in sheep for glutamatergic inputs to KNDy neurons from non-KNDy afferents that are as numerous (luteal phase) or more numerous (LH surge) than glutamatergic input from KNDy neurons (Merkley et al., J Neuroendo 27: 624, 2015).

We thank the reviewer for bringing this work to our attention. We have revised the discussion (L531-539) to include evidence from Merkley *et al.* and how it affects our conclusions.

[Editors' note: further revisions were suggested prior to acceptance, as described below.]

Essential revisions:One of the reviewers has asked for a few additional points for clarification and some speculation by you on the mechanism of altered neurotransmission which might aid in formulating tests of your model.1) The model is explained much more clearly – but there are a few further simplifications that might be made. For example n1,n2, n3 and n4 are all just Hill coefficients and are all = 2; so what seem to be four parameters are just a single constant. Kv,1=Kv,2 – replace with a single parameter? It's just a bit harder than it need be to work out exactly what the model structure is.

We agree that these changes will simplify exposition of the model and we have incorporated them in our manuscript.

2) The manuscript would be improved if the authors were to say whether they think that what they call changes in signaling strength reflect changes in vesicle content, vesicle availability for release, mechanisms coupling activity to vesicle release, receptor availability or post-receptor signaling. I am left uncertain what experiments might be proposed to really test this model (ie potentially refute it), but if any of these were specified then designing a critical test would be simpler.

We have revised the Discussion section to discuss possible mechanisms through which signalling strength and network excitability could be modulated based on recent transcriptomic data (Qiu et al. 2018).